# Pathological Role of High Sugar in Mitochondrial Respiratory Chain Defect-Augmented Mitochondrial Stress

**DOI:** 10.3390/biology13080639

**Published:** 2024-08-21

**Authors:** Ebrima D. Cham, Tsung-I Peng, Mei-Jie Jou

**Affiliations:** 1Department of Physiology and Pharmacology, College of Medicine, Chang Gung University, 259 Wenhua 1st Road, Kweishan, Taoyuan 333, Taiwan; ebrima.cham@rocketmail.com; 2Department of Neurology, Chang Gung Memorial Hospital, Keelung Branch, Keelung 204, Taiwan; tipeng@cgmh.org.tw; 3Department of Medicine, Chang Gung University, Taoyuan 333, Taiwan

**Keywords:** mitochondrial permeability transition pore (MPTP), high glucose toxicity, oxidative stress, mROS, cardiolipin, mitochondrial DNA (mtDNA), mitochondrial calcium stress [Ca^2+^]_m_, complex V

## Abstract

**Simple Summary:**

High glucose levels lead to the production of reactive oxygen species, which, according to some research, is the link between high glucose and the toxicity observed at cellular levels. At normal physiological levels, mitochondrial calcium serves as a second messenger; however, mitochondrial calcium overload leads to the production of reactive oxygen species, which has lethal effects on the cells. The mitochondria have complexes, identified as I, II, III, IV, and V, and any anomaly to these complexes also leads to reactive oxygen species production. Complex V is the enzyme responsible for catalyzing the final step of oxidative phosphorylation by coupling the translocation of protons in the mitochondria for adenosine triphosphate (ATP) synthesis to take place. Mitochondrial complex V T8993G mutation blocks the translocation of protons, hence blocking the production of ATP, leading to a condition called Neuropathy, Ataxia, and Retinitis Pigmentosa (NARP) syndrome. In this study, we explore the possibility of mitochondrial calcium overload mediating the toxicity of high glucose levels in defective respiratory chain-mediated mitochondrial stress. The NARP cells harbor 98% of mitochondrial DNA T8993G mutations, and 143B osteosarcoma cell lines were the parental normal cells used for comparison. Using MTT assays and confocal microscopic techniques, we observed that NARP cells enable reactive oxygen species formation and enhance the depolarization of the mitochondrial membrane potential.

**Abstract:**

According to many research groups, high glucose induces the overproduction of superoxide anions, with reactive oxygen species (ROS) generally being considered the link between high glucose levels and the toxicity seen at cellular levels. Respiratory complex anomalies can lead to the production of ROS. Calcium [Ca^2+^] at physiological levels serves as a second messenger in many physiological functions. Accordingly, mitochondrial calcium [Ca^2+^]_m_ overload leads to ROS production, which can be lethal to the mitochondria through various mechanisms. F1F0-ATPase (ATP synthase or complex V) is the enzyme responsible for catalyzing the final step of oxidative phosphorylation. This is achieved by F1F0-ATPase coupling the translocation of protons in the mitochondrial intermembrane space and shuttling them to the mitochondrial matrix for ATP synthesis to take place. Mitochondrial complex V T8993G mutation specifically blocks the translocation of protons across the intermembrane space, thereby blocking ATP synthesis and, in turn, leading to Neuropathy, Ataxia, and Retinitis Pigmentosa (NARP) syndrome. This study seeks to explore the possibility of [Ca^2+^]_m_ overload mediating the pathological roles of high glucose in defective respiratory chain-mediated mitochondrial stress. NARP cybrids are the in vitro experimental models of cells with F1FO-ATPase defects, with these cells harboring 98% of mtDNA T8993G mutations. Their counterparts, 143B osteosarcoma cell lines, are the parental cell lines used for comparison. We observed that NARP cells mediated and enhanced the death of cells (apoptosis) when incubated with hydrogen peroxide (H_2_O_2_) and high glucose, as depicted using the MTT assay of cell viability. Furthermore, using fluorescence probe-coupled laser scanning confocal imaging microscopy, NARP cells were found to significantly enable mitochondrial reactive oxygen species (mROS) formation and enhance the depolarization of the mitochondrial membrane potential (ΔΨm). Elucidating the mechanisms of sugar-enhanced toxicity on the mitochondria may, in the future, help to alleviate the symptoms of patients with NARP syndromes and other neurodegenerative diseases.

## 1. Introduction

The mitochondria are ancient organelles within eukaryotic cells that have played a pivotal role in evolution by generating ATP through respiration. Mutations in the genes within either nuclear or mitochondrial DNA, which are responsible for encoding proteins essential for aerobic ATP production, can cause a range of human mitochondrial diseases, regardless of the affected organ. In diabetes mellitus, complications involving the vascular organs and multiple organ systems are strongly linked to the excessive production of reactive oxygen species (ROS) triggered by high blood sugar levels [1,2,3]. Additionally, critically ill patients often present with levels of glucose in the blood that are high enough to aggravate multiorgan injuries [4]. High-glucose conditions induce metabolic substrate entry into the mitochondria, and this weakens the electron transport system, culminating in the overproduction of ROS. A deficiency of mitochondrial protein synthesis has been observed in the drosophila mutant tko25t, characterized by respiratory and oxidative phosphorylation defects that lead to developmental delay and sensitivity to seizures as a result of mechanical stress. It was revealed that these mutant effects are aggravated by high dietary sugar intake in a dose-dependent fashion. Series of metabolic abnormalities have also been found as a result of high sugar intake, and some of the lethal effects include decreased NADPH and ATP production and an increase in the levels of lactate and pyruvate [5]. The glycation of extracellular proteins ensues when there is an abnormally high concentration of glucose, such as in cases of diabetic hyperglycemia [6]. Under high-glucose conditions, there is an abnormally high concentration of glucose in the neurons because the absorption of glucose by neurons is insulin-independent. In such instances, glucose is oxidized to form reactive carbonyls and ROS, ultimately activating MAP kinases, and this has effects on the phenotype of the cells [7]. Calcium (Ca^2+^) plays a critical role in mediating many important biological functions and has been implicated as an intracellular regulatory factor in many physiological and pathological processes in the cell. The disruption of intracellular Ca^2+^ homeostasis is frequently associated with the early development of cell injury. Research has established the pathological mechanisms by which intracellular Ca^2+^ overload triggers either necrotic or apoptotic cell death. From studies on different tissues in a variety of pathological conditions, a general consensus has emerged with regard to the role of mitochondrial Ca^2+^ overload as a pivotal link between cellular alterations and mitochondrial dysfunction [8]. Mitochondrial Ca^2+^ uptake is driven by mitochondrial membrane potential (∆Ψm) [9]. Under conditions of oxidative stress, mitochondrial Ca^2+^ cycling can reach critical levels, leading to increased energy expenditure and a dramatic fall in ∆Ψm. Recent work has shown that a fall in mitochondrial ∆Ψm is an early event in apoptosis [10]. Researchers presume that mitochondrial Ca^2+^ uptake is dependent on the membrane potential and the intracellular distribution of the organelle, both of which may be altered in mitochondrial diseases [11]. Mitochondrial permeability transition (MPT) is the key mechanism by which the cell apoptotic pathway is activated. The MPT pore is a mega channel complex that contains an adenine nucleotide translocase (ANT), cyclophilin-D (CyP-D), and a voltage-dependent anion channel (VDAC). The complexity of the pore structure demonstrates how MPT can act as a multifaceted sensor for various cellular messages that might originate from both intra- and extramitochondrial environments and could include Ca^2+^ overload, membrane potential depolarization, oxidative pressure, and receptors connected to cellular signals [12]. The channel complex opening occurs as a result of the binding of CyP-D to ANT in the inner mitochondrial membrane. CyP-D binding enhances the ability of the ANT to undergo a conformational change triggered by Ca^2+^. The binding of ADP or ATP to a matrix site of the ANT antagonizes this effect of Ca^2+^. Modification of other ANT thiol groups inhibits ADP binding and sensitizes the MPT to Ca^2+^ [13]. Increased membrane potential changes the ANT conformation to enhance ATP binding and hence inhibit the MPT [14]. In isolated mitochondria, the opening of the MPT pore leads to the collapse of the electrochemical gradient of H^+^, i.e., causing mitochondrial membrane potential depolarization, thereby annihilating the driving force needed for ATP production and triggering the production of ROS. Moreover, the pore opening also releases several apoptotic proteins, such as cytochrome c, apoptosis-inducing factors (AIFs), and procaspase-9, and turns on cellular apoptotic cascades that eventually lead the cell to the “point of no return” journey of the apoptotic pathway. The MPT is regulated by a variety of cellular physiological and pathological effectors such as mitochondrial Ca^2+^ overload transients, oxidative stress, and depolarization of mitochondrial membrane potential in the concentration of polyamines [15]. The voltage and Ca^2+^ threshold at which MTP pore opening occurs are modulated by a variety of agents and conditions [16]. Cyclosporin A and its non-immunosuppressive analog N methyl-Val-4 cyclosporin A (PKF220-384) both inhibit opening and prevent the translocation of cyclophilin D from the matrix to the membranes of cortical mitochondria [17]. Although the existence of a transient or flicker opening of MPT has long been debated from different aspects, MPT pore flicker has recently been elegantly demonstrated using isolated mitochondria. Mitochondria immobilized on coverslips were imaged using tetramethylrhodamine methyl ester as a membrane-potential indicator. Then, pore opening was triggered by the generation of mROS upon the photodecomposition of the indicator. Pore flicker was apparent as a transient depolarization inhibited by cyclosporin A. This phenomenon has been suggested as a protective mechanism against mitochondrial Ca^2+^ overload and mitochondrial membrane potential depolarization [18]. It is not uncommon to find a decrease in mitochondrial functions in neurodegenerative diseases, which is critical as far as mitochondrial biology is concerned; however, the molecular basis to confirm this is elusive. The human mtDNA is a double-membrane circular molecule of 16,569 bp and it critically encodes 13 polypeptides of the mitochondrial respiratory chain for oxidative phosphorylation, 22 tRNAs, and two small and large subunits of rRNA for protein synthesis. F1F0-ATPase (ATP synthase or complex V) is the enzyme that is responsible for catalyzing the final step of oxidative phosphorylation. This is achieved by F1F0-ATPase coupling the translocation of the protons that are in the intermembrane space and shuttling them to the mitochondrial matrix for ATP synthesis to take place. In mammals, the ATP synthase structure consists of a soluble portion, denoted F1, in which catalyzing ATP synthesis takes place. It also has the F0 portion that is found embedded in the inner membrane of mitochondria and serves as a proton channel. Proton shuttling in F0 takes place at the interface between the ATP6 subunit and the C-ring and enhances conformational changes or modifications that are transmitted to the F1 portion, providing energy for the synthesis of ATP via rotation of the stalk [19]. Mitochondrial DNA (mtDNA)-encoded ATP6 subunit mutations cause complex disorders with varying and heterogeneous expressions and severity, which range from adult-onset Neuropathy, Ataxia, and Retinitis Pigmentosa (NARP) syndrome to a fatal infantile subacute necrotizing encephalomyelopathy, a maternally inherited form of Leigh Syndrome (MILS). The most frequent and first mutation that is associated with NARP/MILS is the ATP6 mutation (T8993G), which results in the substitution of the highly conserved amino acid leucine for arginine (L156R) [20]. NARP cybrids can be established when human osteosarcoma 143B cells are depleted and mtDNA is eliminated by exposing the cells to ethidium bromide, which results in a cell line that is devoid of mtDNA termed rho zero (ρ^0^) cells. Human skin fibroblasts that are obtained from a patient who is clinically confirmed to have NARP syndrome are enucleated and fused with the ρ^0^ cells to create 98% mutant-type (NARP) cybrids [21]. To the best of our knowledge, the effects of high glucose toxicity on NARP cells have not been explored. Our research group hereby hypothesizes that mitochondrial calcium [Ca^2+^]_m_ overload modulates high glucose toxicity on respiratory chain defects in NARP cells. The ultimate aim of this research is to investigate how mitochondrial calcium [Ca^2+^]_m_ overload in tandem with oxidative stress modulates high glucose toxicity on respiratory chain defect-augmented mitochondrial reactive oxygen species (ROS) production, membrane potential (ΔΨ) depolarization, cardiolipin (CL) remodeling, calcium (mCa^2+^) homeostasis, and the mitochondrial permeability transition pore (MPTP), including transient (t-MPT) and permanent MPT (p-MPT).

## 2. Materials and Methods

### 2.1. NARP Cells (Cybrids) and Parental 143B Cells

Both the NARP cybrids and 143B cells described above were kindly provided by our collaborator, Dr. Tanaka, from Japan. These cells had been used in previous studies conducted by our group. The NARP cybrids and rho zero (ρ^0^) cells were maintained in Dulbecco’s modified Eagle’s medium (DMEM) containing 10% fetal bovine serum supplemented with high glucose (4.5 g/L), pyruvate (0.11 mg/mL), and uridine (0.1 mg/mL). NARP cybrids with a high mutant-mtDNA-to-wildtype-mtDNA ratio of approximately 98% were used for experiments, and comparisons were made with the parental osteosarcoma 143B cell line.

### 2.2. Measurement of Cell Viability Using the MTT Assay

Working concentrations of 1 mM, 10 mM, 50 mM, 100 mM, and 500 mM were diluted from a 10 M hydrogen peroxide stock solution. We then exposed the cells to varying stress conditions of hydrogen peroxide at 1 mM, 10 mM, 50 mM, 100 mM, and 500 mM, respectively, for 30 min at room temperature. The cells were then cleaned with HEPES and incubated with MTT solutions for 40 min at 37 °C. Next, cell viability was measured using the ELISA plate reader. Cell viability was detected using the colorimetric 3-(4,5dimethyl-2 thiazolyl)-2,5-diphenyl-2 Htetrazolium bromide (MTT) assay. The activity of the mitochondrial reductase to convert soluble tetrazolium salt into an insoluble formazan precipitate was measured using an enzyme-linked immunoabsorbent assay (ELISA) reader (A-5082; TECAN, Grödig/Salzburg, Austria). The MTT assay was performed 30 min after stress exposure (for hydrogen peroxide) and 48–72 h after high glucose exposure. The activity of the mitochondrial reductase was calculated as the amount of MTT dye conversion in the treated cells relative to that of the sham-treated control cells.

### 2.3. Preparation of the Cells for Imaging

For imaging detection, the cells were grown in a medium consisting of DMEM containing 10% fetal bovine serum supplemented with glucose (4.5 g/L), pyruvate (0.11 mg/mL), and uridine (0.1 mg/mL). All cells were plated onto #1 glass coverslips for fluorescent microscopy.

### 2.4. Chemical and Fluorescent Dye Loading for Fluorescence Measurement of Mitochondrial Events

The loading conditions for each specific fluorescent probe are described as follows: ΔΨm was detected using 200 nM of tetramethylrhodamine methyl ester (TMRM); mCa^2+^ was detected using 2 µM of Rhod-2 AM (Rhod-2); ROS were detected using 2 µM of 6 carboxy2′,7′dichlorodihydrofluorescein diacetate (DCFH-DA); and cardiolipin was detected using 80 nM of nonylacridine orange (NAO). All fluorescent probes were loaded at RT for 30 min, except TMRM, which was loaded for 10 min to prevent quenching. After loading, the cells were rinsed 3 times with HEPES-buffered saline solution (containing 140 mM of NaCl, 5.4 mM of KCl, 1.8 mM of CaCl_2_, 0.8 mM of MgCl, 10 mM of glucose, and 10 mM of HEPES; pH 7.4). The dye-loaded cells were then mounted in a cell chamber for conventional or laser-coupled imaging microscopic observation. 

A total of 7.36 mM of mitoQ stock solution was diluted to the working concentration of 100 µM and aliquoted into 1 mL tubes. Then, 1 µL was pipetted from the working solution to pretreat the cells in the medium for 30 min. Next, another 1 µL was pipetted into 1 mL of HEPES and mixed thoroughly. The cells were stained together with the dyes and then mounted on the confocal microscope for imaging.

An amount of 1 µL from a 10 mM NAO stock solution was pipetted into 1 mL of HEPES solution and the cells were stained for 30 min to detect CL. From a working solution of 1 mM, 1 µL was pipetted into a 1 mL tube containing HEPES to stain the cells for 10 min. The cells were then mounted on the confocal microscope for imaging.

To conduct the staining procedure for MPTP imaging, we used 0.7 µL of CoCl_2_ and 0.5 µL of calcein and mixed in a 1 mL tube of HEPES. Then, the slides were stained for 30 min. Next, an amount of 0.3 µL of TMRM was stained for 10 min. After that, we mounted the slides for confocal imaging.

To precisely control calcium concentrations, we used HEPES-buffered saline containing 140 mM of NaCl, 5.4 mM of KCl, 1.8 mM of CaCl_2_, 0.8 mM of MgCl_2_, 10 mM of glucose, and 10 mM of HEPES at pH 7.4. To make Ca^2+^-free HEPES, we did not add 1.8 mM of CaCl_2_ to the final mixture.

### 2.5. Imaging Analysis of Living Cells

Confocal fluorescence images and image stacks in our lab were collected using a Zeiss LSM 510 META NLO mounted on an Axiovert 200 M inverted microscope (Carl Zeiss Microimaging, Inc., Thornwood, NY, USA). All fluorescence images were collected using a Zeiss objective lens (Plan-Apochromat 100×, NA1.4 oil DIC M27) (Carl Zeiss Microscopy, LLC One North Broadway, Floor 15, White Plains, NY 10601 USA). NAO excitation was conducted using the Argon/2 laser (30 mW). The excitation wavelength was 488 nm, the main dichroic beam splitter was 488/561 nm, and the emission detection filter was bandpass 500–550 nm. All images were processed and analyzed using MetaMorph software, version 7.8 (Universal Imaging Corp., West Chester, PA, USA). Intensity levels were analyzed from the original images and graphed using Microsoft Excel software (Office 2003; Microsoft Corp., Redmond, WA, USA) and Photoshop (version 7.0; Adobe Systems, SanJose, CA, USA). To analyze the mROS and mitochondrial NO (mNO) fluorescence intensity, we selected and measured the regions overlapping with DCFH-DA (to measure ROS) and TMRM (to measure ΔΨm) signals and DAF-FM (to measure NO) and TMRM signals, respectively, thus ensuring that the analyzed regions were indeed mitochondrial.

### 2.6. Statistical Analysis

Data are presented as the means ± standard error (SE) of at least three independent experiments. We used paired *t*-tests between the groups, and a *p*-value of <0.05 was considered significant (*).

## 3. Results

### 3.1. Assessing Cell Viability under Different Stress Conditions

Working concentrations of 1 mM, 10 mM, 50 mM, 100 mM, and 500 mM were diluted from a 10 M hydrogen peroxide stock solution. We then exposed the cells to varying stress conditions of hydrogen peroxide at 1 mM, 10 mM, 50 mM, 100 mM, and 500 mM for 30 min at room temperature. Next, the cells were cleaned with HEPES and incubated with MTT solutions for 40 min at 37 °C. Then, the cell viability was measured using the ELISA plate reader. As expected, the NARP cybrids exhibited significant cell toxicity compared with the normal parental control 143B cells (Figure 1A). Furthermore, we tested and treated the cells with glucose at varying concentrations of 25 mM (control), 50 mM, and 75 mM. The cells were exposed for 48 h, after which cell viability was determined using the MTT assay. As expected, a dose-dependent toxicity of glucose on the cells was observed, and significantly greater toxicity was found in the NARP cells compared with the parental 143B cells (Figure 1B), indicating that NARP enhances toxicity.

### 3.2. Determining the Toxicity of High Glucose and Mitochondrial Calcium [Ca^2+^]_m_ Overload on 143B Cells and NARP Cybrids 

To determine the toxicity of high glucose and mitochondrial calcium [Ca^2+^]_m_ overload on 143B cells and NARP cybrids, both the 143B cells and NARP cybrids were prepared and incubated overnight with Dulbecco’s modified Eagle’s medium supplemented with either low glucose (25 mM) or high glucose (75 mM). To induce oxidative stress, we transferred and stored the cells in either HEPES containing Ca^2+^- or Ca^2+^-free HEPES, and we then mounted the cells on a Zeiss LSM 510 META NLO confocal microscope with a 561 nm lens. The microscope images were analyzed to determine the effects or toxicity caused by high glucose and mitochondrial calcium [Ca^2+^]_m_ overload on membrane potential depolarization levels measured using TMRM (red) stain and oxidative stress levels measured using DCF (green). As expected, the NARP cybrids depolarized faster with greater dose-dependent toxicity in the Ca^2+^-treated group, with a corresponding increase in DCF fluorescence intensity (Figure 2A,B). The 143B cells also showed dose-dependent toxicity in both TMRM depletion and DCF levels. The Ca^2+^-free cells depolarized much slower, with a minimal rise in mROS levels (DCF) compared with the Ca^2+^-treated cells (Figure 2C,D). Therefore, the results demonstrate that the higher the glucose concentrations, the greater the toxicity and ROS levels produced under Ca^2+^-treated conditions, indicating that complex V mutation under high-glucose conditions enhances mitochondrial stress.

### 3.3. Depicting How Mitochondrial Calcium Overload [Ca^2+^]_m_ Modulates High Glucose Toxicity on Respiratory Chain Defect-Augmented Mitochondrial Reactive Oxygen Species (ROS) Production and Membrane Potential (ΔΨ) Depolarization in Parental 143B Cells and NARP Cybrids

In attempting to assess how mitochondrial calcium overload [Ca^2+^]_m_ modulates high glucose toxicity on respiratory chain defect-augmented mitochondrial reactive oxygen species (ROS) production and membrane potential (ΔΨ) depolarization, we prepared two solutions: a HEPES solution containing calcium chloride (CaCl_2_) and a HEPES solution without CaCl_2_ (Ca^2+^-free). 143B and NARP cells were incubated with either low glucose (25 mM) or high glucose (75 mM) for 24 h. The cells were stained with fluorophores with either Ca^2+^-containing HEPES or Ca^2+^-free HEPES and mounted on the LSM 510 META NLO confocal microscope to observe the effects. The cells were treated in two pairs or groups. The 143B cells were treated with either 25 mM of glucose or 75 mM of glucose and the NARP cells were treated with either 25 mM of glucose or 75 mM of glucose. One group was stained in either Ca^2+^-containing HEPES or Ca^2+^-free HEPES solution, and comparisons were made between the two groups after they were irradiated with 561 nm laser irradiation to assess the membrane potential depolarization and ROS effects on the two cell types at different concentrations of glucose. Laser irradiation at 561 nm was pointed at three different spots on an individual cell, namely the irradiation point (IR), the near irradiation point (NEAR), and far from the irradiation point (FAR). The membrane potential was analyzed and assessed for depolarization and ROS formation. Laser iterations at 561 nm started at 20 s and ended at 300 s. We discovered that the Ca^2+^-treated cells for both the 143B and NARP cybrids showed faster depolarization of the membrane potential than did the Ca^2+^-free cells. The high-glucose-treated NARP cells showed the fastest depolarization and highest DCF levels compared with the 143B cells; moreover, the Ca^2+^-treated cells showed significant levels of toxicity (Figure 3B–D). Based on these observations, we postulate that at high glucose concentrations (75 mM), mitochondrial Ca^2+^ overload augments cellular toxicity in both the NARP cells and 143B cells up to this point, with the NARP cells showing significantly higher toxicity than the 143B cells; however, the mechanisms involved are yet to be determined. We concluded that NARP-induced F1F0-ATPase inhibition augments mitochondrial toxicity, especially in the presence of Ca^2+^ and high glucose.

### 3.4. Depicting How Mitochondrial Calcium Overload [Ca^2+^]_m_ Modulates High Glucose Toxicity on Respiratory Chain Defect-Augmented Mitochondrial Reactive Oxygen Species (ROS) Production and Membrane Potential (ΔΨ) Depolarization in Parental 143B Cells and NARP Cybrids under Antioxidant Conditions 

The results presented in Figure 2 and Figure 3 demonstrate that calcium notably contributes to the toxicity observed under high-glucose conditions. Hence, we then explored how mitochondrial calcium overload [Ca^2+^]_m_ modulates high glucose toxicity on the respiratory chain defect-augmented mitochondrial reactive oxygen species (ROS) production and membrane potential (ΔΨ) depolarization in parental 143B cells and NARP cybrids under antioxidant conditions. Mitoquinone (mitoQ) is a potent mitochondria-targeted antioxidant synthesized by covalent attachment of a Coenzyme Q10 (quinone) to a lipophilic triphenylphosphonium cation to allow selective mitochondria targeting for the prevention of mitochondrial reactive oxygen species (mROS). The mitochondrial membrane potential (Δψm) allows the cations to accumulate selectively in mitochondria up to 1000-fold compared with non-targeted Coenzyme Q. Once produced, mROS indiscriminately damage mitochondrial components and, more importantly, they crucially directly activate the mitochondrial permeability transition (MPT), which is a critical mechanism in the initiation of post-mitochondrial apoptotic signaling. No significant depolarization of the mitochondrial membrane potential or oxidative stress levels between the Ca^2+^-treated and Ca^2+^-free cells was observed (Figure 4). This result could indicate that ROS scavenging protects the mitochondrial membrane potential and maintains the production of mROS to sub-lethal levels; moreover, the production of mROS may be critical for Ca^2+^ to mediate high glucose toxicity. In conclusion, we postulate that mitoQ has the potential to rescue the high glucose toxicity in both the NARP and 143B cells observed above by scavenging mROS. 

### 3.5. Demonstrating How Mitochondrial Calcium Overload Modulates High Glucose Toxicity on Respiratory Chain Defect-Augmented Mitochondrial Cardiolipin (CL) Remodeling and Calcium (Ca^2+^) Homeostasis in Parental 143B Cells and NARP Cybrids under Ca^2+^-Treated Hepex Conditions

Cardiolipin (CL) is a unique phospholipid found in the inner mitochondrial membranes that give mitochondria its unique architectural structure and function. According to a clinical and experimental model, malfunctioning or damaged CL could lead to abysmal mitochondrial function, which might be linked to a host of diseases. Oxidation levels also have a direct link to many disease types, including metabolic disorders and neuronal dysfunction, as well as cardiomyopathy. We hypothesized that high-glucose conditions could overwhelm CL functions and remodeling. Comparable levels of CL were detected almost throughout the time-lapse recordings up to 600 s in both the 25 mM glucose-treated 143B cells (Figure 5A) and NARP cells (Figure 5C). Rhod-2 fluorescence intensity levels were also found to be comparable after testing both the 143B cells and NARP cells with low glucose (25 mM) (Figure 5A,C). CL levels were depleted from 100% down to 68% for IR and 84% for the NEAR and FAR points. However, there was an increase in Rhod-2 levels for the IR, NEAR, and FAR points at 80%, 60%, and 45%, respectively, in parental 143B cells after high-glucose (75 mM) treatments (Figure 5B). NAO/CL fluorescence intensity levels were lower following testing with high glucose (75 mM), as the NAO fluorescence intensity levels started to deplete steeply after just 20 s and mitochondrial calcium (mCa^2+^) homeostasis levels almost doubled, respectively, at IR 100%, NEAR 83%, and FAR 93% in 143B cells (Figure 5D). With this array of mitochondrial events, we determined that under high-glucose conditions, NARP induces the inhibition of F1F0-ATPase, augments CL remodeling via depletion, and increases mCa^2+^ homeostatic events, which could have lethal consequences on the cells as CL, mCa^2+^, and mPTP are interconnected. Cardiolipin was detected using 80 nM of nonylacridine orange (NAO, green) and mCa^2+^ using 2 µM of Rhod-2 AM (Rhod-2, red) at 850 nm irradiations.

### 3.6. Investigation of How High Glucose Augments Respiratory Chain Defect-Augmented Mitochondrial Permeability Transition Pore (MPTP), Including Transient (t-MPT) and Permanent (p-MPT), on Parental 143B Cells and NARP Cybrids

The mitochondrial permeability transition pore is a channel that is composed of voltage-dependent anion channels that span the mitochondrial outer membrane. The channel serves the physiological function of sensing environmental oxidative stresses, including Ca^2+^, amongst other ions and compounds. This is achieved through the flicker opening of the pore holes. However, this could overwhelm the cell functions if the pore is continuously or permanently kept open, often leading to apoptosis. After incubating the cells with high and low glucose and staining, we mounted the cells on a Zeiss LSM 510 META NLO confocal microscope for imaging. In order to determine the transient or permanent opening of the mPTP, we coloaded the cells with TMRM and calcein in the presence of Co^2+^ (for mPT). The appearance of a transient tMPT opening is indicated by the loss of calcein and maintenance of TMRM signals, while the simultaneous loss of both calcein and TMRM fluorescence signals is indicative of the permanent mPTP. We observed that transient t-MPT could be detected under low glucose (25 mM) conditions in the parental 143B cells as can be seen by the depletion of the calcein stains at 720 s (Figure 6A, panel I) compared with the TMRM stains at the same time point (Figure 6A, panel II). The same trend was found for the fluorescence intensity (FI) levels shown in the graphs. After 640 s, the fluorescence intensity for calcein falls steeply (Figure 6A, panel III) compared with the TMRM fluorescence intensity (Figure 6A, panel IV). Permanent MPT (p-MPT) might have been demonstrated in parental 143B cells under high-glucose (75 mM) conditions as the TMRM stains start to fade at 620 s in the time-lapse recordings and almost completely fade at 660 s (Figure 6B, panel II); moreover, the calcein stains also started fading at the same time points (Figure 6B, panel I). After testing the NARP cells with 25 mM of glucose, the calcein stains almost completely faded away after 920 s (Figure 6C, panel I) compared with the TMRM stains, which persisted until the end of the time-lapse recording (Figure 6C, panel II), albeit fading slightly. Permanent MPT (p-MPT) may have been depicted in the NARP cells after testing with high glucose as the TMRM and calcein levels depleted after just 220 s in the time-lapse recordings (Figure 6D, panels I and II). As mitochondrial CL, mCa^2+^, and MPTP are all linked and interconnected, and CL maintains the cell’s ultrastructure, it is unsurprising that t-MPT is depicted in both the 143B and NARP cells under low-glucose conditions (Figure 6A and Figure 6C, respectively), just the same way as p-MPT is detected in both NARP and 143B cells under high-glucose conditions (Figure 6B and Figure 6D, respectively). It is worth noting that p-MPT was achieved much faster in the NARP cells, at just 220 s, compared with the 143B cells, which started much later at 620 s. Therefore, these results confirm that the mitochondrial complex V mutation enhanced high-glucose stress on mitochondria.

## 4. Discussion 

In this study, we sought to explore the possibility of mitochondrial calcium overload [Ca^2+^]_m_ mediating the pathological roles of high glucose in defective respiratory chain-mediated mitochondrial stress. In line with our hypothesis, we first determined the viability of the cells to be used in the experiment. As expected, the NARP cells harboring mitochondrial DNA point mutation T8993G show significantly greater dose-dependent toxicity under both hydrogen peroxide (H_2_O_2_) (Figure 1A) and high-glucose conditions (Figure 1B). We then put into perspective our main hypothesis that Ca^2+^ mediates glucose toxicity. Both the 143B and NARP cells were used under the same experimental conditions of Ca^2+^-treated and Ca^2+^-free HEPES solutions for low and high glucose. Treating the cells with Ca^2+^ induced significant membrane potential depolarization and ROS levels in NARP cybrids compared with Ca^2+^-free treatments, which did not exhibit statistically significant differences (Figure 2). Additionally, we combined imaging and graphical representation to delineate the effects of mitochondrial Ca^2+^ overload mediating high glucose toxicity in the 143B and NARP cells. Consistent with the findings in Figure 2, Ca^2+^ treatment induced toxicity and faster membrane potential depolarization in the Ca^2+^-treated NARP cells, especially under high-glucose conditions (Figure 3). Since it has been reported that ROS are the link between high glucose and the toxicity observed at cellular levels [22], we sought to scavenge ROS and determine whether Ca^2+^ can independently induce the toxicity observed in Figure 3. To scavenge ROS production, we used a powerful antioxidant, mitoquinone (mitoQ), which is a potent mitochondria-targeted antioxidant synthesized by covalent attachment of a Coenzyme Q10 (quinone) to a lipophilic triphenylphosphonium cation. Mitoquinone prevents oxidative damage as it is a selective mitochondria-targeted antioxidant that is lipophilic in nature, carrying a positive charge allowing it to accumulate in the matrix space by crossing the cell membranes. Once in the matrix, it faces the surface of the mitochondrial inner membrane and is well positioned to reduce ROS production [23]. Interestingly, by employing this technique, Ca^2+^ could not independently induce any significant changes in both TMRM and DCF levels in both the 143B cells and NARP cells, which indicates that for Ca^2+^ to induce any significant toxicity, ROS presence is a critical factor. We also discovered that mitochondrial calcium overload leads to the depletion of CL, which is an important component of the cell ultrastructure to maintain integrity [24]. Furthermore, we showed that the higher the glucose levels, the higher the calcium homeostasis, which leads to an increase in Rhod-2 levels in the NARP cells compared with the 143B cells. The mitochondrial permeability transition pore (MPTP) is a part of the normal physiological response or process of the cell whereby chemicals, solutes, and compounds can be easily flushed through the channel opening. This is achieved through the transient or flicker opening. We hypothesized that mitochondrial Ca^2+^ overload could overwhelm this opening and create a permanent opening. In this state, the opening becomes permanent and the cell contents are released unchecked, which ultimately leads to the apoptosis of the cell. In line with our hypothesis, we demonstrated that mitochondrial Ca^2+^ overload led to the depletion of the calcein stains significantly faster than the TMRM stains under low glucose conditions in both the 143B and NARP cybrids (Figure 6A,C). Conversely, we may have detected a permanent opening of the pores when the cells were treated with high-glucose conditions, as evidenced when the TMRM and calcein stains both started to deplete at the same time points in the 143B and NARP cells (Figure 6B,D). To the best of our knowledge, this is the first time that the effects of high glucose toxicity on the augmentation of mitochondrial stress via NARP-induced F1F0-ATPase inhibition, as well as the protective actions of mitoQ in scavenging ROS production and protecting the mitochondrial membrane potential, have been tested.

The mechanism behind the enhancement of mROS formation following high-glucose treatment via NARP-induced F1F0-ATPase inhibition is yet to be determined. Previous studies have suggested that mROS formation is enhanced as a consequence of F1F0-ATPase inhibition and mitochondrial coupling, owing to the hyperpolarized membrane potential in NARP cells leading to mitochondrial respiratory chain decreased activity [25,26]. In one of our recent studies, mROS formation was found to be potentiated in NARP cells upon the introduction of apoptotic insults such as laser irradiation-induced ROS stress and amyloid beta treatment [27]. Moreover, our group also previously reported that mROS formation was augmented by a defect of mtDNA, resulting in apoptosis in common deletion (mtDNA 4977 bp deleted) cybrids [28,29], with the same phenomenon occurring in RBA-1 astrocytes that harbor mitochondrial complex I defects, owing to long term rotenone exposure. This result demonstrates that several neurodegenerative disorders could be potentiated due to complex defects or mitochondrial DNA mutations. Additionally, we previously reported that CL is an important pathological target for mitochondrial apoptotic insults such as ionomycin and arachidonic acid hydrogen peroxide in NARP cybrids [30]. The results of this study establish that in NARP-induced inhibition of F1F0-ATPase, CL is a crucial pathological target for high-glucose-induced overproduction of mROS and mCa^2+^ overload. Moreover, our results confirm that NARP-induced overproduction of mROS leads to greater depletion of CL in NARP cells compared with 143B cells. It is generally known that CL depletion or peroxidation ensues as a result of mROS overproduction, which leads to a depressed respiratory chain or impaired mitochondrial function [31,32,33]. Mitochondrial respiratory chain impairment and CL depletion or peroxidation, as well as mROS overproduction, are interconnected to create a synergistic circle, culminating in a mitochondrial bioenergetics crisis producing low ATP output and mitochondrial impairment [34]. Severely depleted or peroxidized CL can enhance the opening of the mPTP, which could meet the minimum threshold for Ca^2+^ and thus further enhance and potentiate mCa^2+^ toxicity in the opening of the mPTP. Cytochrome C release from the mitochondria is the ultimate effect of peroxidized CL on mitochondria [35,36].

MitoQ is a potent antioxidant as it can concentrate in the mitochondria 1000-fold. MitoQ delays aging and protects the heart, brain, and liver from oxidative damage [37]. Mitochondrial damage is a healthcare concern that can be found in a number of diseases, including Alzheimer’s disease and liver and heart conditions, and it is also an observed phenomenon usually associated with hair loss, osteoporosis, and aging [38]. In this study, mitoQ showed great potential and promise in that it effectively scavenged mROS, helped maintain membrane potential, and blocked high-glucose-induced mCa^2+^ toxicity in both NARP and 143B cells. Another research group reported that when catalase removed free radicals from the mitochondria, a 10% increment in the lifespan of the mouse was recorded, and they projected that similar results would be achieved in humans; however, there are currently no clinical trials to support this claim [39]. Despite these breakthrough findings, there are also potential mitoQ limitations. Coenzyme Q10 (CoQ_10_) is another mitochondrial antioxidant that is not easily absorbed by the body, with a significant amount of it not reaching the mitochondria when taken as supplements. MitoQ might serve as a remedy in this situation; however, it is not as powerful as CoQ_10_. Additionally, mitoQ does not serve as a substitute for CoQ_10_’s critical role in generating energy [40]. Due to these limitations, it would be essential to find alternatives to mitoQ, probably melatonin, alpha-tocopherol, or ubiquinol. Furthermore, another limitation of this study is that we used Ca^2+^-free HEPES solutions and HEPES containing Ca^2+^ as our source of Ca^2+^. In a future study, it would be worthwhile to use specific mCa^2+^ channel blockers, such as ruthenium red, to block mitochondrial Ca^2+^ uptake. The results of this study show that NARP-induced inhibition of F1FO-ATPase augments mROS overproduction and mCa^2+^ stress under high-glucose conditions and ultimately leads to mitochondrial apoptotic death, which is in line with our hypothesis. Moreover, this study establishes that mitoQ successfully scavenges mROS overproduction in 143B cells and, to some extent, NARP cells, maintaining and rescuing the membrane potential.

This study demonstrates, for the first time, that mitoQ rescues apoptotic cell death in high glucose toxicity in NARP cells. It has been reported that ROS play a central role in the pathologies of many diseases, chiefly type 2 diabetes mellitus and neurodegenerative diseases. Diabetes induces the production of ROS, which has pathological effects on various cellular components, affecting homeostasis and cellular health in general. The effects of ROS are multifaceted and include, but are not limited to, compromised insulin secretion, beta cell dysfunction, insulin resistance, multiple signaling pathways, and exacerbated neurodegenerative conditions in the body [41,42]. MitoQ could potentially alleviate these pathological mROS insults and high glucose toxicity in patients with diabetes mellitus, NARP syndromes, and neuroinflammation by scavenging mROS.

## 5. Conclusions

Based on the data presented in this study, mitochondrial DNA T8993G mutation (NARP) syndrome induces greater toxicity in NARP cells compared with 143B cells. Moreover, high glucose toxicity is modulated by mitochondrial calcium overload [Ca^2+^]_m_, and the presence of ROS is critical.

## Figures and Tables

**Figure 1 biology-13-00639-f001:**
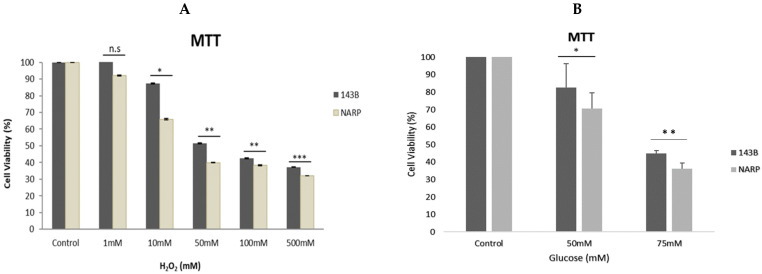
Assessing the cell viability of 143B cells and NARP cybrids at various concentrations [toxicity] of (**A**) hydrogen peroxide [H_2_O_2_] and (**B**) glucose, demonstrating the dose-dependent toxicity of H_2_O_2_ and glucose in MTT assays in (**A**) and (**B**), respectively. All experiments were performed in biological triplicate; results are given as the mean and standard error, and significance was determined using pairwise *t*-tests where * *p* < 0.05, ** *p* < 0.01, and *** *p* < 0.001. n.s denotes non-significance.

**Figure 2 biology-13-00639-f002:**
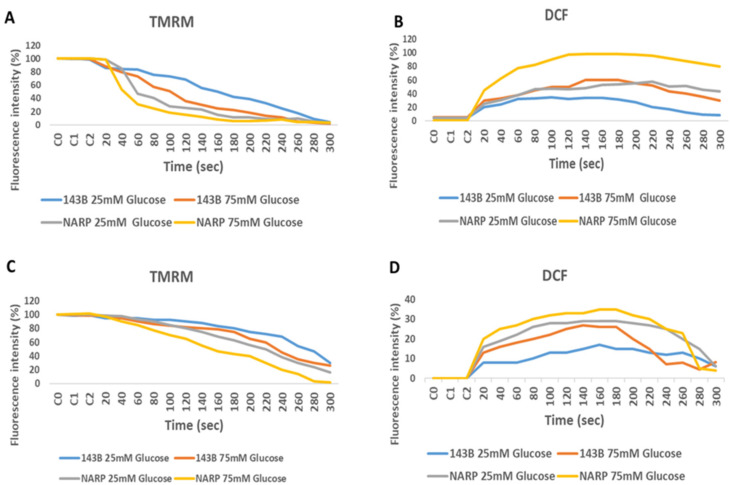
Assessing the effects of high and low glucose in the presence or absence of Ca^2+^ between 143B cells and NARP cybrids. 143B and NARP cells were incubated with high or low glucose solutions and treated with either Ca^2+^-containing HEPES or one without Ca^2+^. The Ca^2+^ group showed greater toxicity, as shown by the faster TMRM depletion and increased mROS production levels shown in (**A**) and (**B**), respectively, compared with the Ca^2+^-free HEPES-treated cells shown in (**C**) and (**D**), respectively.

**Figure 3 biology-13-00639-f003:**
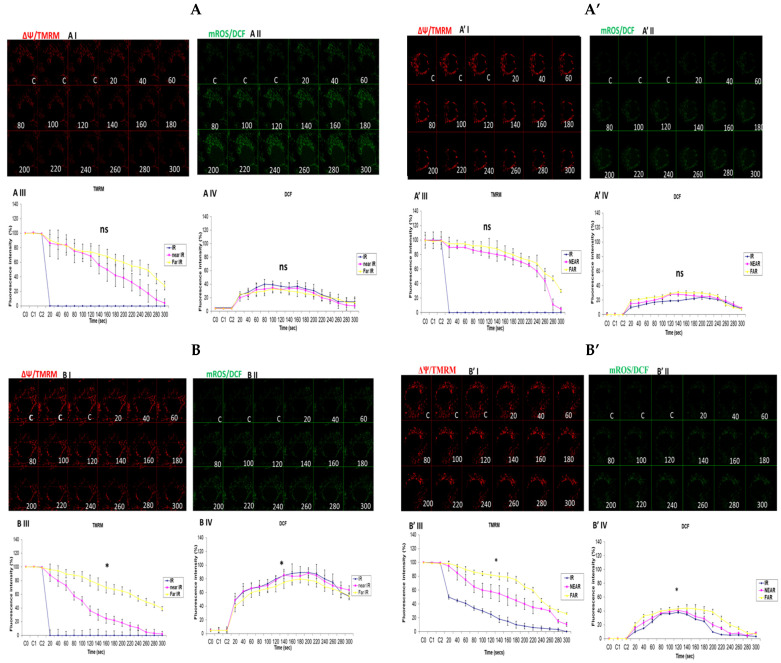
(**A**,**A’**) 143B cells: (**A**) 25 mM glucose- and Ca^2+^-treated cells; (**A’**) 25 mM glucose-treated, Ca^2+^-free cells. The Ca^2+^-treated 143B cells showed comparable levels of membrane potential depolarization (TMRM) and oxidative stress levels (DCF) to those of the Ca^2+^-free cells. (**B**,**B’**) High glucose toxicity on 143B cells: both cell types were high-glucose-treated (75 mM), with (**B**) Ca^2+^-treated and (**B’**) Ca^2+^-free groups. The Ca^2+^-treated 143B cells depolarized significantly faster, with significantly higher oxidative stress (DCF) levels than the Ca^2+^-free 143B cells. (**C**,**C’**) NARP cells: (**C**) 25 mM glucose- and Ca^2+^-treated cells and (**C’**) 25 mM glucose-treated Ca^2+^-free cells. The Ca^2+^-treated NARP cells depolarized significantly faster with higher DCF values than the Ca^2+^-free cells. (**D**,**D’**) NARP cells: (**D**) 75 mM glucose- and Ca^2+^-treated cells and (**D’**) 75 mM glucose-treated, Ca^2+^-free cells. The Ca^2+^-treated NARP cells depolarized significantly faster, with significantly higher oxidative stress levels than the Ca^2+^-free high-glucose-treated cells. All panels labeled I, II, III, and IV show TMRM (red), oxidative stress DCF (green) levels, and graphical representations (fluorescence intensity %) of TMRM and DCF levels, respectively. All experiments were performed in biological triplicate; results are given as the mean and standard error, and significance was determined using pairwise *t*-tests (* *p* < 0.05, and ** *p* < 0.01). ns denotes non-significance.

**Figure 4 biology-13-00639-f004:**
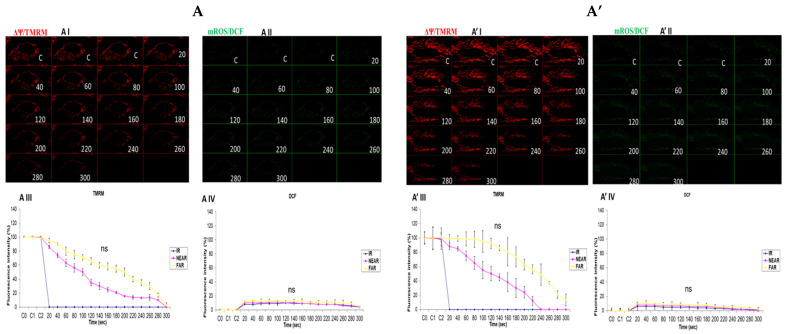
(**A**,**A’**) 143B cells: (**A**) 25 mM glucose-, Ca^2+^-, and MitoQ-treated cells and (**A’**) 25 mM glucose- and MitoQ-treated but Ca^2+^-free cells. The Ca^2+^-treated 143B cells had comparable levels of membrane potential depolarization (TMRM, red) and oxidative stress (DCF, green) levels with the 25 mM glucose- and MitoQ-treated but Ca^2+^-free cells. (**B**,**B’**) 143B cells: (**B**) 75 mM glucose-, Ca^2+^-, and MitoQ-treated cells and (**B’**) 75 mM glucose- and MitoQ-treated but Ca^2+^-free cells. The Ca^2+^-treated cells depolarized relatively faster, with comparable levels of DCF with the Ca^2+^-free cells. (**C**,**C’**) NARP cells: (**C**) 25 mM glucose-, Ca^2+^-, and MitoQ-treated cells and (**C’**) 25 mM glucose- and MitoQ-treated but Ca^2+^-free cells. The Ca^2+^-treated NARP cells depolarized relatively faster, with comparable DCF levels with the Ca^2+^-free 25 mM glucose-treated cells. (**D**,**D’**) NARP cells: (**D**) 75 mM glucose-, Ca^2+^-, and MitoQ-treated cells and (**D’**) 75 mM glucose- and MitoQ-treated but Ca^2+^-free cells. The Ca^2+^-treated NARP cells had comparable levels of membrane potential depolarization (TMRM, red) and ROS production levels with the Ca^2+^-free 75 mM glucose-treated cells. All panels labeled I, II, III, and IV show TMRM (red), oxidative stress DCF (green) levels, and graphical representations (fluorescence intensity %) of TMRM and DCF levels, respectively. ns denotes statistical non-significance.

**Figure 5 biology-13-00639-f005:**
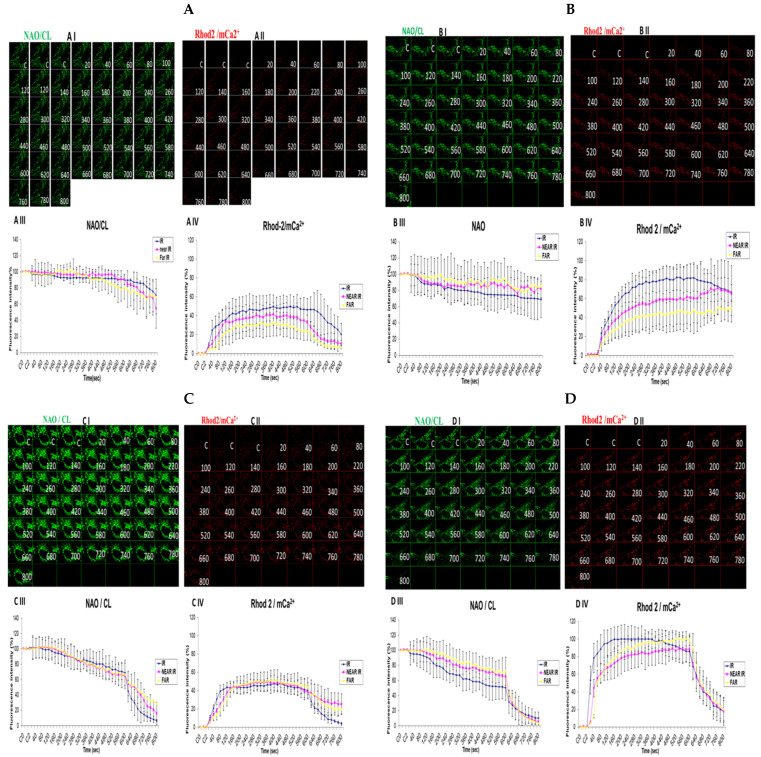
Demonstrations of how high sugar augments respiratory chain defect-augmented mitochondrial cardiolipin (CL) remodeling and calcium (Ca^2+^) homeostasis in parental 143B cells and NARP cybrids. The toxicity of high glucose on CL and mCa^2+^ homeostasis was compared as follows: 143B cells using (**A**) 25 mM of glucose and (**B**) 75 mM of glucose; NARP cells using (**C**) 25 mM of glucose and (**D**) 75 mM of glucose. All panels labeled I, II, III, and IV show NAO/CL (green), Rhod-2/mCa^2+^ (red), and graphical representations (fluorescence intensity %) of NAO/CL and Rhod-2 levels, respectively.

**Figure 6 biology-13-00639-f006:**
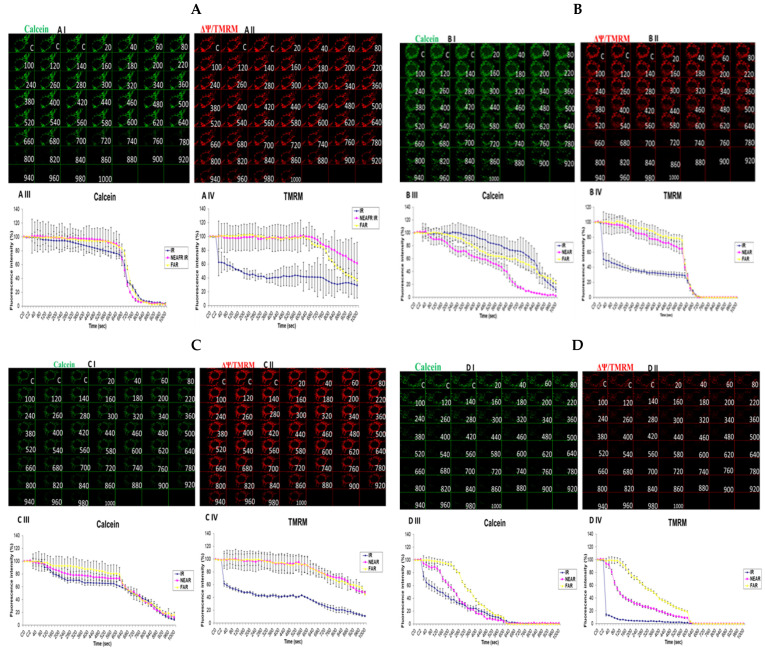
Investigation of how high glucose augments respiratory chain defect-augmented mitochondrial permeability transition pore (MPTP), including transient (t-MPT) and permanent (p-MPT), on parental 143B cells and NARP cybrids. Investigation of p-MPT and t-MPT in 143B cells and NARP cybrids showing t-MPT under low glucose conditions in both (**A**) 143B cells and (**C**) NARP cells, and p-MPT under high-glucose conditions in (**B**) 143B cells and (**D**) NARP cells. All panels labeled I, II, III, and IV show calcein (green), TMRM (red), and graphical representations (fluorescence intensity %) of calcein and TMRM levels, respectively.

## Data Availability

Data are contained within the article and are available upon request to the corresponding author.

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
