# Peer review of "Pathological Role of High Sugar in Mitochondrial Respiratory Chain Defect-Augmented Mitochondrial Stress"

_biology, 2024, doi:10.3390/biology13080639_

Round 1

Reviewer 1 Report

Comments and Suggestions for Authors

With a lot of interest I carefully read the manuscript entitled "Pathological role of high sugar in mitochondrial respiratory chain defect - augmented mitochondrial stress" by Cham et al. The authors use MTT assays and confocal microscopy to study the link between calcium overload and toxicity of high glucose in NARP cybrid cells.

Main issues:

The results section mostly describes the methods, not the results. The entire discussion is actually the text of the results section and the discussion itself is missing. It reads like a PhD thesis, not like a manuscript. The authors do not discuss their results. 

Also, the figures look sloppy. Different sizes left versus right panel, figure title on top of the figure frame, etc. Fig. 1A says MTT, fig 1B MTT assay. Fig 1A ctrl, 1B control. Although this is all very easy to fix, it's concerning. If the authors can't be bothered producing decent figures, it raises doubts about the execution of the experiments.

One experiment is spread over two pages. Authors should present the main conclusion of the experiment and show the details in supplementary figures. 

Comments on the Quality of English Language

The English language is acceptable. 

Reviewer 2 Report

Comments and Suggestions for Authors

In this manuscript, the authors have demonstrated that cells harboring mitochondrial defect (coxV mutation) are more sensitive towards high glucose conditions, leading to calcium overload, increased ros and apoptosis. The findings are interesting but unfortunately, the manuscript is not well written and in most cases the image quality is too low and the figure legends are so small that the data cannot be deciphered properly. Also, a lot of sentences are very repetitive. The authors can improve their manuscript by working on formatting, English sentences and provide higher quality images with proper figure legends. Also conclusion is required at the end of each result section. My specific comments are below:

1) Figure 1: The concentrations in the result section does not match the data.

Please maintain same formatting.

The authors can quantify data as percentage change for better data visualization.

2) Figure 2: Axis legends are too small. 

Figure 2A III: Since the axis font is too small, it's difficult to understand, but it seems that the colors are reversed.

Instead of making so many separate graphs, the authors can club the 4 plots together eg. 143B 25mM, 143B 75mM, NARP 25mM, NARP 75mM. That way the 4 samples can be visualized together.

3) Figure 3: Image and plot quality are too low. I cannot read the data. How is the conclusion of figure 3 different from Figure 2? The authors need to write a conclusion sentence after each of their result section.

4) Figure 4: Conclusion statement missing. Also Mito-Q prep should be in the methods section, not here.

Image quality too low.

5) Figure 5 & 6: Preparation of dyes should be in the methods section and conclusion statement is missing. Once again, image and plot quality too low.

Comments on the Quality of English Language

Quality of English language requires significant improvement. The manuscript has too many spelling mistakes, redundant and repetitive sentences.

Reviewer 3 Report

Comments and Suggestions for Authors

High glucose induces excessive production of reactive oxygen species (ROS), linking it to cellular toxicity, exacerbated by respiratory complex anomalies. Mitochondrial calcium overload due to high glucose can lead to ROS generation, potentially damaging mitochondria. The study explores these mechanisms using NARP cybrids, bearing the T8993G mutation in mitochondrial complex V, showing increased apoptosis, mitochondrial ROS formation, and membrane potential depolarization under hydrogen peroxide and high glucose conditions.

The study investigates cell viability under various stress conditions induced by hydrogen peroxide (H2O2) and glucose in both 143B cells and NARP cybrids. Hydrogen peroxide at concentrations ranging from 1mM to 100mM and glucose at 25mM, 50mM, and 75mM were applied to cells for specified durations. Results showed significant toxicity in NARP cybrids compared to 143B cells, particularly under oxidative stress conditions. Imaging and assays revealed pronounced membrane potential depolarization and oxidative stress levels in NARP cybrids, highlighting their heightened susceptibility to metabolic and oxidative stressors.

Major concerns and experimental suggestions

Clarify the mechanism of action of mito-Q and discuss potential limitations or alternative interpretations of the results.

the method description should include details on how calcium concentrations were precisely controlled and measured throughout the experiments. Variability in calcium levels could significantly impact the observed outcomes, especially in comparisons between different cell types and treatments.

Use pharmacological inhibitors of MCU (e.g., Ruthenium 360) to block mitochondrial calcium uptake during hydrogen peroxide exposure. Measure ROS levels using DCF assay and monitor membrane potential with TMRM staining under different conditions.

Investigate how varying glucose concentrations affect mitochondrial function and oxidative stress responses. Assess mitochondrial ROS production and membrane potential using DCF and TMRM assays respectively, under hydrogen peroxide exposure conditions.

Round 2

Reviewer 1 Report

Comments and Suggestions for Authors

The authors improved the paper. No further comments

Comments on the Quality of English Language

The manuscript would benefit from some language improvement to help the flow

Reviewer 2 Report

Comments and Suggestions for Authors

The authors have significantly improved the manuscript. The result sections now have a conclusion statement and there is a logical flow between the results. The work itself is scientifically sound and will be of importance to the scientific community. I understand that the authors want to put the figures side-by-side which compromises the image quality, but even though the work is nice, it will be difficult for the readers to decipher the data as some of the images and plots still lack quality and the axis labels are too small.

Reviewer 3 Report

Comments and Suggestions for Authors

authors have addressed all the concerns and manuscript can now be considered for publication
